

# Identification of high-efficiency 3′GG gRNA motifs in indexed FASTA files with ngg2

Elisha D. Roberson

Departments of Medicine & Genetics, Division of Rheumatology, Washington University in Saint Louis, Saint Louis, MO, United States of America

## ABSTRACT

CRISPR/Cas9 is emerging as one of the most-used methods of genome modification in organisms ranging from bacteria to human cells. However, the efficiency of editing varies tremendously site-to-site. A recent report identified a novel motif, called the 3′GG motif, which substantially increases the efficiency of editing at all sites tested in *C. elegans*. Furthermore, they highlighted that previously published gRNAs with high editing efficiency also had this motif. I designed a Python command-line tool, ngg2, to identify 3′GG gRNA sites from indexed FASTA files. As a proof-of-concept, I screened for these motifs in six model genomes: *Saccharomyces cerevisiae*, *Caenorhabditis elegans*, *Drosophila melanogaster*, *Danio rerio*, *Mus musculus*, and *Homo sapiens*. I also scanned the genomes of pig (*Sus scrofa*) and African elephant (*Loxodonta africana*) to demonstrate the utility in non-model organisms. I identified more than 60 million single match 3′GG motifs in these genomes. Greater than 61% of all protein coding genes in the reference genomes had at least one unique 3′GG gRNA site overlapping an exon. In particular, more than 96% of mouse and 93% of human protein coding genes have at least one unique, overlapping 3′GG gRNA. These identified sites can be used as a starting point in gRNA selection, and the ngg2 tool provides an important ability to identify 3′GG editing sites in any species with an available genome sequence.

## INTRODUCTION

Genome engineering allows for the targeted deletion or modification by homology directed repair of a target locus. Currently, one of the most popular methods for genome manipulation is the clustered regularly interspaced short palindromic repeat (CRISPR)/CRISPR associated protein 9 (Cas9) system adapted from *Streptococcus pyogenes*. The *S. pyogenes* CRISPR/Cas system was initially thought to represent a novel DNA repair mechanism, but was eventually found to provide heritable bacterial immunity to invading exogenous DNA, such as plasmids and bacteriophages (*Barrangou et al., 2007*; *Makarova et al., 2006*). During endogenous CRISPR/Cas9 function, foreign DNA integrates into the CRISPR locus. The bacterial cell then expresses the pre-CRISPR RNA (crRNA) and a trans-activating crRNA (tracrRNA) that pair to form a complex that

Corresponding author
Elisha D. Roberson,
eroberso@dom.wustl.edu

is cleaved by RNAse III (*Deltcheva et al., 2011*). The resulting RNA is a hybrid of the pre-crRNA and the tracrRNA, and includes a 20 bp guide RNA (gRNA) sequence. The gRNA is incorporated into Cas9 and can then guide the cleavage of a complementary DNA sequence by the nuclease activity of the Cas9 protein. The topic of CRISPR-Cas genome editing has been reviewed extensively elsewhere (*Doudna & Charpentier, 2014*; *Hsu, Lander & Zhang, 2014*; *Jiang & Doudna, 2015*; *Mali, Esvelt & Church, 2013*).

Codon-optimized versions of Cas9 are available for a wide range of organisms, and can easily be synthesized if it is not already available. Transfecting cells with Cas9 plasmid along with a fused crRNA-tracrRNA hybrid construct called a single-guide RNA (sgRNA) allows for temporary activity of Cas9. Alternatively, cells can also be transfected with a Cas9 protein preloaded with a gRNA to reduce off target effects (*Kim et al., 2014*). Keeping a stock of plasmids with a sgRNA backbone minus the gRNA site makes it easy to quickly generate new sgRNA plasmids by site-directed mutagenesis. The Cas9 protein loaded with the sgRNA will bind to sites complementary genomic loci, but will only cut it if a protospacer adjacent motif (PAM) site immediately follows the complementary sequence (*Mojica et al., 2009*). The PAM site for the commonly-used *Streptococcus pyogenes* type-II CRISPR is an NGG motif. Therefore, a *S. pyogenes* Cas9 gRNA site can be defined as $N_{20}NGG$. It is important to note that constitutively expressed sgRNAs typically use a U6 snRNA promoter that strongly prefers a G starting base. For U6 compatibility, sequences starting with A, C, or T may be used if they are cloned into a sgRNA vector with an appended G base, resulting in a 21 bp gRNA (*Farboud & Meyer, 2015*; *Ran et al., 2013b*), or by incorporating the gRNA into a tRNA poly-cistron and taking advantage of tRNA processing cleavage (*Xie, Minkenberg & Yang, 2015*). I will refer to the subset gRNA sites contain a starting G base ($GN_{19}NGG$) as canonical 3′GG gRNA sites.

The rate of editing using the CRISPR/Cas9 system is far higher than homologous recombination, but higher efficiency is still desirable. The introduction of a longer stem in part the sgRNA stem-loop structure and the flip of a single A in a polyA track of a separate sgRNA stem-loop, called the flip + extension (F + E) sgRNA design, resulted in increased Cas9 editing efficiency (*Chen et al., 2013*). Recently, another improvement was reported that increases efficiency. gRNA sites with a GG motif adjacent to the PAM site, called 3′GG gRNAs, have far higher activity than equivalent gRNA sites in the same region (*Farboud & Meyer, 2015*). These sites take the form of $N_{18}GGNGG$. The 3′GG motif efficiency in species other than *C. elegans* is unknown.

Tools already exist to identify *S. pyogenes* Cas9 gRNA targets in sequences via a web interface for an input DNA, or for common model organisms (*Gratz et al., 2014*; *Heigwer, Kerr & Boutros, 2014*; *Liu et al., 2015*; *Montague et al., 2014*; *Naito et al., 2015*; *Stemmer et al., 2015*; *Xiao et al., 2014*). However, there are limitations to these methods. Searching a whole genome for gRNA sites is not feasible via a web interface unless the genome is exceptionally small. There is already support for most model organisms, but leaves individuals working on less commonly studied species without a resource. In this manuscript, I report a Python command-line tool, ngg2, for identification of 3′GG gRNA motifs from indexed FASTA genome files. As a proof of concept, I report all 3′GG gRNA

motifs in 6 model species plus two additional mammalian genomes, identifying more than 88 million sites, of which more than 60 million are unique matches within the reference genome for that species. More than 83% of all protein coding genes in 7/8 species have at least one unique 3′GG gRNA overlapping it for potential editing.

## MATERIALS & METHODS

### ngg2 motif identification

I designed ngg2 using Python with compiled regular expressions for the 3′GG gRNA plus PAM motif. The use of compiled regular expressions makes the search quite efficient even for relatively large genomes. This tool is Python based, relying on the Python base functions and some external dependencies, such as the regex and pyfaidx packages. ngg2 uses the FASTA index via pyfaidx (*Shirley et al., 2015*) to directly seek the genomic target without reading the entire file. The default mode scrapes the entire FASTA input for 3′GG gRNA sites, but individual contigs or contig regions can be specified instead. ngg2 identifies these sites on both the sense and antisense strands independently for each chromosome, facilitating multiprocessing to decrease computation time. ngg2 buffers all detected gRNA sites in memory, and then identifies uniqueness by storing the gRNA sites in a dictionary. This means that all unique sites will be appropriately flagged, but near matches, i.e., single-base mismatches will not. The output from this tool could be pipelined with other tools, or further extended with BioPython to allow for identification of near matches as they are beyond the scope of this tool. The output can be extended to include non-canonical sites starting with any base. ngg2 output includes the contig name, start and end positions, the gRNA sequence, the PAM sequence, whether the site starts with a G, and whether the gRNA sequence was unique in the searched region. For a whole-genome this is very handy, but be aware that selecting only a small region will only tell you if a gRNA is unique within the region, not the genome. The source code for ngg2 is available from GitHub.

### Multi-species site identification

I used ngg2 to identify all 3′GG gRNA motifs 6 commonly studied organisms and two others: *Saccharomyces cerevisiae*, *Caenorhabditis elegans*, *Drosophila melanogaster*, *Danio rerio*, *Mus musculus*, *Homo sapiens*, *Sus scrofa*, and *Loxodonta africana*. I used a GNU Make script to download genomes and GTF gene annotations, calculate genome GC content, and annotate genes in R to enable reproducibility. The Makefile downloads the top-level or primary assembly genomes from Ensembl Release 79, runs ngg2 on all contigs for each FASTA file, and calculates GC content for each genome. I based the GC content of each genome from non-N base content.

After identifying gRNA sites, I used R, particularly relying on the plyr, dplyr, tidyr, magrittr, GenomicRanges, and GenomicFeatures packages, to identify the overlap of each gRNA with gene exons and tabulate the number of genes overlapping at least one gRNA (*Lawrence et al., 2013*; *R Core Team, 2014*). A gRNA was considered overlapping a gene if at least one base of gRNA sequence overlapped at least one base of exonic sequence. The best

**Table 1 Count of gRNA classes in each species.** All $N_{18}$GGNGG motifs are included in the 'All gRNAs' section, while only canonical gRNAs starting with a G are in the 'Canonical gRNAs' section. The 'All' class accumulates all matching motifs for that section, while the 'Unique' class counts only sites with on exact match in the reference genome.

| | All gRNAs | | Canonical gRNAs | |
|---|---|---|---|---|
| | All | Unique | All | Unique |
| *S. cerevisiae* | 44,757 | 41,462 | 9,938 | 9,717 |
| *C. elegans* | 379,955 | 333,752 | 85,887 | 82,696 |
| *D. melanogaster* | 929,164 | 815,501 | 243,705 | 238,460 |
| *D. rerio* | 5,815,459 | 3,110,150 | 835,035 | 744,702 |
| *M. musculus* | 19,368,938 | 13,925,626 | 3,856,020 | 3,660,550 |
| *S. scrofa* | 18,711,809 | 12,716,221 | 4,145,116 | 3,558,512 |
| *H. sapiens* | 23,022,656 | 14,782,453 | 4,172,179 | 3,954,608 |
| *L. africana* | 20,276,122 | 14,929,328 | 4,075,522 | 3,893,752 |
| **Total** | **88,548,860** | **60,654,493** | **17,423,402** | **16,142,997** |

case puts the cut site within the exon body and should certainly disrupt the gene. The worst case of a 1bp overlap cutting in an intron should still generate indels big enough to extend into the exon or to delete a canonical splice site. I calculated all summary statistics and generated ggplot2 figures using RStudio (v0.98.1102) Markdown with knitr (*Xie, 2013*).

## RESULTS

### 3′GG gRNA sites are common in each species

Overall, I identified greater than 88 million 3′GG gRNA sites in the tested genomes (Table 1). Some of these gRNA sequences were not unique in a given genome, leaving more than 60 million unique 3′GG sites. Approximately 16 million of the 60 million unique sites were canonical G starting motifs. The sites identified in each species with the gRNA sequence, PAM sequence, genome coordinates, annotated overlapping genes, and number of perfect genome matches are available for download (*Roberson, 2015*). The R scripts, Python files, and Make files are also available in a public repository for reproducibility.

The genomes I analyzed had vastly different sizes, ranging from approximately 12 Mb for yeast to greater than 3 Gb for humans and elephants, and as a result had dramatically different numbers of 3′GG gRNA sites per genome. Therefore, I also assessed the site density per megabase of reference genome size (Table 2). Unique sites with a G starting base averaged a density of 1,218 sites/Mb, or 1 site per 821 bp. All unique sites averaged 4,210 sites/Mb, or 1 unique 3′GG gRNA site per 238 bp. *D. rerio* had the lowest density at 527 unique G-start sites/Mb, while *D. melanogaster* had the highest density at 1,659 unique sites/Mb. The low density of unique sites in zebrafish may be due to genome complexity from previous duplication events

I profiled the performance of canonical G-start gRNA searches in each of the tested genomes for both block and exhaustive scans using both 1 and 10 CPUs (Table 3). The parallelization in this program is by contig and strand, so the maximum utilized number

**Table 2  3′GG gRNA sites per megabase genome size.** Reference genome size was determined from the species FASTA index. The number of unique 3′GG gRNA sites in the genomes is encouraging, with an average across all species of one unique site per kb of genome.

| | All gRNAs | | Canonical gRNAs | |
| --- | --- | --- | --- | --- |
| | **All** | **Unique** | **All** | **Unique** |
| *S. cerevisiae* | 3,681.55 | 3,410.52 | 817.46 | 799.29 |
| *C. elegans* | 3,788.70 | 3,327.99 | 856.42 | 824.60 |
| *D. melanogaster* | 6,464.83 | 5,674.00 | 1,695.62 | 1,659.13 |
| *D. rerio* | 4,117.24 | 2,201.93 | 591.19 | 527.24 |
| *M. musculus* | 7,092.58 | 5,099.33 | 1,412.01 | 1,340.43 |
| *S. scrofa* | 6,662.50 | 4,527.72 | 1,475.90 | 1,267.04 |
| *H. sapiens* | 7,427.26 | 4,768.92 | 1,345.97 | 1,275.78 |
| *L. africana* | 6,342.71 | 4,670.14 | 1,274.89 | 1,218.03 |

**Table 3  Run times with one and multiple CPUs.** Profiling was performed using Python v2.7.3 using 1 or 10 processors on a server with Intel i7-3930K processors and 32 GB of RAM. Canonical gRNAs were searched for benchmark purposes. When possible, it is clearly advantageous to use multiple processors to accelerate gRNA searches.

| | Block | | | Exhaustive | | |
| --- | --- | --- | --- | --- | --- | --- |
| | **1 CPU** | **10 CPU** | **Delta** | **1 CPU** | **10 CPU** | **Delta** |
| *Saccharomyces cerevisiae* | 0.9 | 0.3 | −71% | 1.2 | 0.4 | −68% |
| *Caenorhabditis elegans* | 6.4 | 1.4 | −78% | 8.1 | 2.1 | −74% |
| *Drosophila melanogaster* | 67.8 | 12.7 | −81% | 71.7 | 13.6 | −81% |
| *Danio rerio* | 99.3 | 20.3 | −80% | 138.2 | 26.8 | −81% |
| *Mus musculus* | 186.0 | 47.7 | −74% | 284.1 | 66.6 | −77% |
| *Sus scrofa* | 536.4 | 111.1 | −79% | 633.2 | 126.7 | −80% |
| *Homo sapiens* | 207.4 | 53.9 | −74% | 306.2 | 71.6 | −77% |
| *Loxodonta africana* | 293.4 | 64.8 | −78% | 398.3 | 79.9 | −80% |

of threads would be twice the number of contigs. Using 10 CPUs reduced runtimes by approximately 70–80% in all cases. It is worth noting that exhaustively scraping the human genome for canonical sites took only 71.6 s with 10 CPUs, and even the longest search took only 126.7 s for *Sus scrofa* using 10 CPUs.

## Little strand bias observed for canonical 3′GG gRNA sites

The strand of each gRNA site with respect to the reference was included in the ngg2 output files. For each organism, I considered every gRNA site as an independent Bernoulli trial with a 50% probability of a "Sense" strand designation as a successful trial outcome (Table 4). 5/8 species showed strand bias for all gRNA sites (*C. elegans*, *D. melanogaster*, *D. rerio*, *H. sapiens*, *L. africana*). Only *C. elegans* and *H. sapiens* demonstrated strand bias significantly different from the expected ratio for canonical 3′GG sites. While the difference in strand selection is significant, it may be unimportant to editing site selection. Wildtype Cas9 cleaves both DNA strands simultaneously, and therefore the strand of the target

**Table 4  Strand bias for gRNA sites.** The gRNA type is either all 3′GG sites or only canonical G starting gRNA sites. The estimate column is the estimated rate of positive strand selection observed. The *p*-value column is detected for whether the Bernoulli trial estimates differ significantly a 50/50 strand selection, and the adjusted *p*-value is based on a Benjamini–Hochberg false-discovery rate correction.

| gRNA type | Species | Estimate | *p*. value | *p*. adj |
|---|---|---|---|---|
| All | *Saccharomyces cerevisiae* | 0.500 | 9.02E−01 | 1.00E+00 |
| | *Caenorhabditis elegans* | 0.494 | 9.09E−12 | 1.36E−10 |
| | *Drosophila melanogaster* | 0.498 | 8.86E−06 | 9.75E−05 |
| | *Danio rerio* | 0.501 | 6.22E−04 | 6.22E−03 |
| | *Mus musculus* | 0.500 | 6.52E−01 | 1.00E+00 |
| | *Homo sapiens* | 0.501 | 9.59E−19 | 1.53E−17 |
| | *Loxodonta africana* | 0.499 | 4.02E−06 | 4.83E−05 |
| | *Sus scrofa* | 0.500 | 4.88E−01 | 1.00E+00 |
| Canonical | *Saccharomyces cerevisiae* | 0.501 | 8.00E−01 | 1.00E+00 |
| | *Caenorhabditis elegans* | 0.490 | 1.50E−10 | 2.10E−09 |
| | *Drosophila melanogaster* | 0.500 | 6.09E−01 | 1.00E+00 |
| | *Danio rerio* | 0.501 | 9.30E−02 | 7.44E−01 |
| | *Mus musculus* | 0.500 | 4.57E−02 | 4.11E−01 |
| | *Homo sapiens* | 0.501 | 2.01E−06 | 2.62E−05 |
| | *Loxodonta africana* | 0.500 | 9.11E−01 | 1.00E+00 |
| | *Sus scrofa* | 0.500 | 4.45E−01 | 1.00E+00 |

sequence doesn't matter. Strategies that employ dual nickases to reduce off target effects could be affected by such bias, as they require two separate gRNA sites on opposite strands (*Ran et al., 2013a*). The difference observed is less than 0.6% different from expected 50% ratio, and whether this functionally affects the ability to choose paired 3′GG gRNAs remains to be seen.

## CGG & GGG PAM sites are underrepresented

I visualized the distribution of the four PAM sites (AGG, CGG, GGG, TGG) as a stacked bar chart of each sites proportion of the total identified sites in each species (Fig. 1). In general, the AGG and TGG sites represented the majority of 3′GG gRNA sites in all species. I tested whether PAM site distribution differed from chance based on the GC content of the reference genome. For each species, I considered each PAM site a Bernoulli trial, and defined success as either CGG or GGG site identity. The probability of success was set equal to the estimated genome-wide GC content calculated from the reference genome, excluding N bases (Table 5). None of the tested genomes met the expected GC success rate. The rate of picking a CGG or GGG PAM was less than the genome GC content in *S. cerevisiae*, *M. musculus*, *Sus scrofa*, *Loxodonta africana*, and *H. sapiens*. In particular, the estimate for *M. musculus*, *H. sapiens*, and *Loxodonta africana* was >10% different from the genome GC fraction. This is not necessarily unexpected. The CGG PAM site includes a 5′ CpG dinucleotide that is generally underrepresented due to the relatively high frequency of methyl-cytosine deamination to thymine. *C. elegans*, *D. melanogaster*, and *D. rerio* were the exceptions, with CGG and GGG PAM selection greater than the expected frequency.

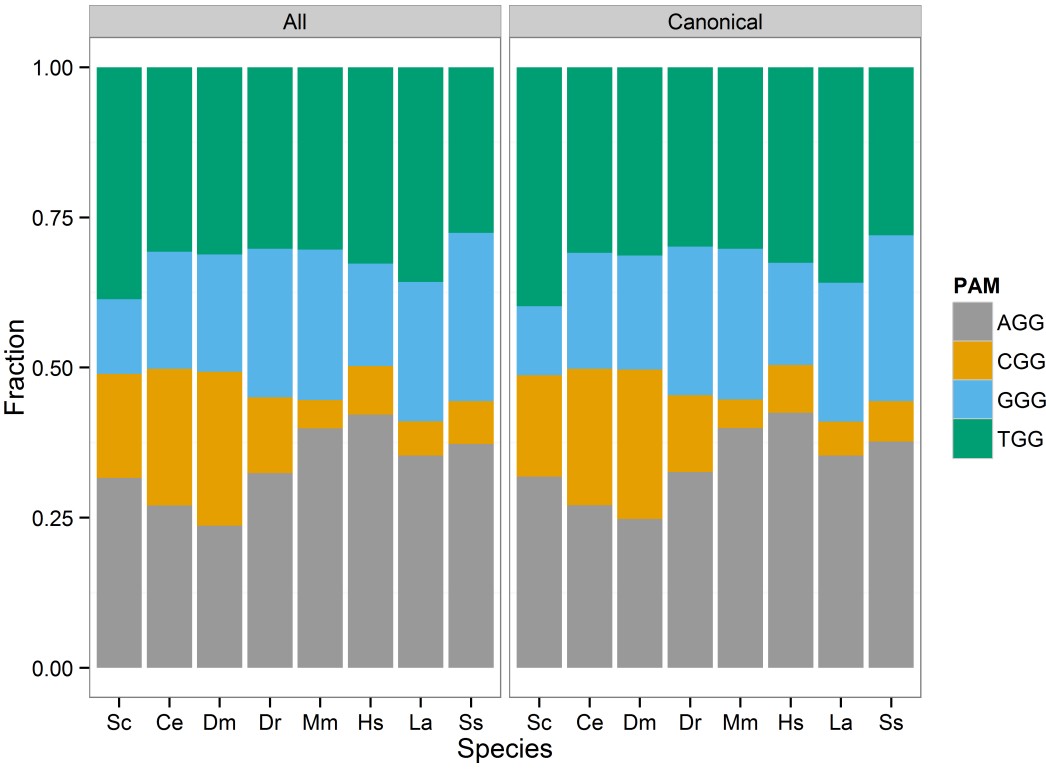

**Figure 1 PAM site usage across tested species.** Each species has four potential protospacer adjacent motifs (PAM) possible for identified gRNA sites. The stacked bar chart shows the fraction of all PAM sites each motif occupies. The CGG motif, that includes a CpG dinucleotide, is the least prevalent motif in the zebrafish, mouse, human, elephant, and pig genomes.

However, *C. elegans* may not be unexpected, as it lacks DNA methylation and would not necessarily be at an advantage to limit CpG dinucleotides.

## Most protein coding genes overlap at least one unique 3′GG gRNA

A common use of genome engineering is to knock out or otherwise modify the function of a protein coding gene. The efficiency of such edits is critical, as just introducing frame-shifting mutations can require screening a large number single-cell clones or derived animals to identify a successful edit. As part of this study, I annotated for each gRNA in the 8 species if there was any overlap with a gene. Conversely, I also annotate a count of how many of each of the four classes (all sites, all unique sites, canonical sites, and unique canonical sites) overlap every gene. No less than 89% of any species' genes overlap at least one unique 3′GG gRNA (Table 6). This catalog of potential sites demonstrates that most protein coding genes can be targeted by at least one 3′GG gRNA site to achieve high editing efficiency.

## DISCUSSION

In this manuscript, I have described a new tool for identifying 3′GG gRNA sites and presented a catalog of potential editing sites in 8 species. Importantly, many genomic loci

**Table 5 PAM site frequency compared to genome GC content.** The average genome GC content and the estimated chance of picking a GC PAM site (CGG or GGG) are shown for each species. GC content was calculated from the downloaded reference files.

| gRNA_type | Species | gc | Estimate | *p*. value | *p*. adj |
|---|---|---|---|---|---|
| All | Saccharomyces cerevisiae | 0.382 | 0.298 | 2.30E−301 | 1.10E−300 |
| | Caenorhabditis elegans | 0.354 | 0.422 | 1.98E−323 | 3.01E−322 |
| | Drosophila melanogaster | 0.420 | 0.452 | 3.46E−323 | 4.79E−322 |
| | Danio rerio | 0.367 | 0.373 | 7.90E−218 | 3.20E−217 |
| | Mus musculus | 0.417 | 0.298 | 1.58E−322 | 1.40E−321 |
| | Homo sapiens | 0.409 | 0.251 | 1.68E−322 | 1.40E−321 |
| | Loxodonta africana | 0.408 | 0.289 | 1.58E−322 | 1.40E−321 |
| | Sus scrofa | 0.417 | 0.352 | 1.58E−322 | 1.40E−321 |
| Canonical | Saccharomyces cerevisiae | 0.382 | 0.284 | 1.00E−98 | 2.10E−98 |
| | Caenorhabditis elegans | 0.354 | 0.420 | 9.88E−324 | 1.58E−322 |
| | Drosophila melanogaster | 0.420 | 0.438 | 4.70E−81 | 4.70E−81 |
| | Danio rerio | 0.367 | 0.376 | 1.60E−102 | 4.80E−102 |
| | Mus musculus | 0.417 | 0.299 | 8.40E−323 | 1.10E−321 |
| | Homo sapiens | 0.409 | 0.250 | 8.40E−323 | 1.10E−321 |
| | Loxodonta africana | 0.408 | 0.288 | 8.40E−323 | 1.10E−321 |
| | Sus scrofa | 0.417 | 0.344 | 8.40E−323 | 1.10E−321 |

**Table 6 Fraction of genes overlapping at least one gRNA.** Ensembl GTF files were used to annotate overlap of gRNA sites with known genes. A gene was called as potentially cut if at least one gRNA overlapped at least 1 base with an exon of that gene. Most genes in the 8 species have at least one unique cut per gene.

| Species | All motifs | | Canonical motifs | |
|---|---|---|---|---|
| | All | Unique | All | Unique |
| S. cerevisiae | 0.93 | 0.90 | 0.65 | 0.62 |
| C. elegans | 0.96 | 0.83 | 0.81 | 0.68 |
| D. melanogaster | 0.99 | 0.97 | 0.91 | 0.89 |
| D. rerio | 0.89 | 0.61 | 0.74 | 0.42 |
| M. musculus | 0.99 | 0.96 | 0.90 | 0.84 |
| S. scrofa | 0.99 | 0.86 | 0.92 | 0.76 |
| H. sapiens | 0.98 | 0.93 | 0.92 | 0.84 |
| L. africana | 0.91 | 0.87 | 0.61 | 0.59 |

can be targeted by unique 3′GG gRNA sites. The efficiency of 3′GG gRNA sites in species other than *C. elegans* has yet to be established, but is worth further study. This tool reports the uniqueness of identified sites, but blast searching of potential gRNA sequences is warranted to identify near-match sites. It is also important to consider the target genome's specific genotypes when designing a gRNA. In particular, variants that alter PAM sites away from NGG will not be cleaved by Cas9 even if the gRNA is an exact match.

The accuracy of editing can be improved by using two gRNAs and a mutant Cas9 nickase. I observed significant, but low-effect strand bias in these genomes. This may lead to some loci not being compatible with paired 3′GG gRNA sites. When possible, choosing paired 3′GG gRNA sites should be strongly considered. Efficiencies of less than 10% were increased to 50% efficiency or greater by using the 3′GG strategy (*Farboud & Meyer, 2015*). As such, using paired 3′GG gRNAs with a nickase may give the best of both worlds with both high accuracy and high efficiency.

It is important to note that ngg2 will operate on any indexed FASTA file. Many gRNA site finding tools are limited to catalogs of gRNA sites in model organisms. This tool fills an important gap for individuals working outside of commonly used species, demonstrated by the use of ngg2 on the genomes of *S. scrofa* and *L. africana*. The provided gRNA site survey and associated tool, ngg2, represent a valuable resource for designing genomic modification strategies.

## ACKNOWLEDGEMENTS

I wish to thank Dr. Li Cao for her helpful comments during the preparation of this manuscript, and Dr. Matthew Shirley for his suggested use of pyfaidx.

### Funding

A portion of effort spent on designing this software was supported under NIH P30 AR048335 as an activity of the Human Genomics and Bioinformatics Facility in the Washington University Rheumatic Disease Core Center. The funders had no role in study design, data collection and analysis, decision to publish, or preparation of the manuscript.

### Grant Disclosures

The following grant information was disclosed by the author:
NIH: P30 AR048335.

### Competing Interests

I have no competing interests related to this manuscript or tool.

### Author Contributions

- Elisha D. Roberson conceived and designed the experiments, performed the experiments, analyzed the data, contributed reagents/materials/analysis tools, wrote the paper, prepared figures and/or tables, performed the computation work, reviewed drafts of the paper.

### Data Availability

https://github.com/RobersonLab/ngg2

https://github.com/RobersonLab/2015_ngg2_manuscript

http://dx.doi.org/10.6084/m9.figshare.1515944.

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
