# Peer review of "Identification of high-efficiency 3′GG gRNA motifs in indexed FASTA files with ngg2"

_PeerJ Computer Science, doi:10.7717/peerj-cs.33_

## Round 0.1 · original submission · Major Revisions

Dear Dr. Roberson,

After reading your work and considering the comments from reviewers, we agree that the work is of value to the community, however, we would like to see some additional experiments to support your hypothesis. Please address the reviewer's comments and provide the additional experimental results and discussion, or rebuttal if we are requesting work that is beyond the scope of this submission.

Reviewer 1 ·

Basic reporting

No comments.

Experimental design

Lines 20 and 68-69. The Farboud & Meyer paper only worked with nematodes so it is not yet clear the 3’GG motif will be as effective in other systems.

Lines 40-46 . The commonly used CRISPR/Cas system, i.e. Cas9, is a Type II system. The Types I and III have different components.

Line 54-55. It is not correct to say “the PAM site for a Type I CRISPR is an NGG”. First Cas9 is Type II, not Type I. Second, Type II CRISPR systems (Cas9) are found in different species of bacteria, and the Cas9 protein in different species has evolved different PAM motif requirements. The commonly used S. pyogenes Cas9 has the NGG preference. Other species, i.e. N. meningitidis, use different PAMs. (GATT in that species).

Line 85. Wouldn't it be pretty easy to upgrade the tool to report uniqueness of the gRNA? at least report yes or no?

Line 87-88 and 170. BLAT is not very useful for checking off-targets, since off-targets can contain mismatches and BLAT does not work well for detecting CRISPR targets with even one or a few mismatches. They are too short and BLAT was designed to detect near-perfect matches of this length or longer. BLAST is appropriate as long as the parameters are set to allow mismatches to a certain degree within the target.

Line 160. How was gene overlap defined - is this within the coding regions of exons, or merely within the entire transcription unit? In other words does the tool return gRNA sites in both exons and introns of the gene? This was not clear at all, but it is very important, as large genes may have many intronic matches but few exonic matches and I am not sure how this was performed. This needs to be made very explicit.

Validity of the findings

No comments

Additional comments

This computational tool is fine and conceptually very simple. I agree that is may be of good use to researchers who are dealing with model organism sequences/genomes for which CRISPR targets have not been pre-cataloged.

I do think it may be premature to regard the “3’GG” motif as a guarantee of high efficiency targets, as the Farboud & Meyer paper only involved nematodes. This motif has not yet been shown to be consistently valuable in other organisms or systems. Although it may well be, I don’t think the Farboud paper really well explained how this motif was shown to be efficient in previous papers, although they do make this claim. Therefore I believe the author should introduce this note of caution into the manuscript.

Lines 20 and 68-69. The Farboud & Meyer paper only worked with nematodes so it is not yet clear the 3’GG motif will be as effective in other systems.

Lines 40-46 . The commonly used CRISPR/Cas system, i.e. Cas9, is a Type II system. The Types I and III have different components.

Line 54-55. It is not correct to say “the PAM site for a Type I CRISPR is an NGG”. First Cas9 is Type II, not Type I. Second, Type II CRISPR systems (Cas9) are found in different species of bacteria, and the Cas9 protein in different species has evolved different PAM motif requirements. The commonly used S. pyogenes Cas9 has the NGG preference. Other species, i.e. N. meningitidis, use different PAMs. (GATT in that species).

Line 85. Wouldn't it be pretty easy to upgrade the tool to report uniqueness of the gRNA? at least report yes or no?

Line 87-88 and 170. BLAT is not very useful for checking off-targets, since off-targets can contain mismatches and BLAT does not work well for detecting CRISPR targets with even one or a few mismatches. They are too short and BLAT was designed to detect near-perfect matches of this length or longer. BLAST is appropriate as long as the parameters are set to allow mismatches to a certain degree within the target.

Line 160. How was gene overlap defined - is this within the coding regions of exons, or merely within the entire transcription unit? In other words does the tool return gRNA sites in both exons and introns of the gene? This was not clear at all, but it is very important, as large genes may have many intronic matches but few exonic matches and I am not sure how this was performed. This needs to be made very explicit.

Reviewer 2 ·

Basic reporting

No comments

Experimental design

This work to identify unique 3'GG motif sites to use in targeted genome editing is of clear utility and value to the community. The ngg2 tool, if improved to address the issues below, would be a good contribution in this area.

1. Comparison to existing tools
- lacks specific references, tool descriptions
- uninformative explanation of "less common, non-model" organism issue which defines the novelty of the current work (ngg2)
- no clear evidence / discussion of how ngg2 provably addresses issues with non-model genomes where others did not

2. Uniqueness of sites
- not clear why ngg2 omits this, given the emphasized critical importance of identifying unique sites? This step is more work than the what ngg2 currently performs
- manuscript includes results for uniqueness, possibly using blast/blat; no details / parameters provided
- genome sequence is already available and of interest in most applications, and redundant sites are easily collated (hash, map data structures)
- since no results are presented for the "allowN" option, very difficult to gauge its utility and impact on uniqueness and runtime

3. [Python] code design / implementation / evaluation
- relies on FASTA sequence data, previously indexed in a specific (samtools) format
- an undo limitation: (a) requires tool/pre-processing dependencies, and (b) does not allow pipelining
- simply using community library (biopython) or even a discussion of why this was not done, would be expected
- no runtime results/discussion presented; important given application / evaluation on whole genomes

Validity of the findings

How would 3'GG sites disrupt a protein-coding gene if they are located are in introns/UTRs? Needs clearer description of "overlap" criteria

Additional comments

One additional minor suggestion for the manuscript : Figures 1 and 3 seem extraneous and less informative versions of the data in tables - might move Suppl. Table 3 to main text as it is an important use-case for using the tool

---

## Round 0.2 · accepted · Accept

You clearly addressed all of the reviewer's concerns. You also made the necessary modifications to the software and manuscript. The scientific value of the paper is clearer and hopefully this is a useful tool for the community.